# Clinical Characteristics and Outcomes of Patients with Acute Respiratory Failure Due to SARS-CoV-2 Interstitial Pneumonia Treated with CPAP in a Medical Intermediate Care Setting: A Retrospective Observational Study on Comparison of Four Waves

**DOI:** 10.3390/jcm12041562

**Published:** 2023-02-16

**Authors:** Silvia Accordino, Ciro Canetta, Greta Bettini, Federica Corsico, Gabriele Ghigliazza, Laura Barbetta, Christian Folli, Valeria Savojardo, Francesco Blasi

**Affiliations:** 1High Care Internal Medicine Unit, Internal Medicine Department, Foundation IRCCS Ca’ Granda Ospedale Maggiore Policlinico, via Francesco Sforza 35, 20122 Milan, Italy; 2Pulmonology and Cystic Fibrosis Unit, Internal Medicine Department, Foundation IRCCS Ca’ Granda Ospedale Maggiore Policlinico, via Francesco Sforza 35, 20122 Milan, Italy; 3Department of Pathophysiology and Transplantation, University of Milan, via Francesco Sforza 35, 20122 Milan, Italy

**Keywords:** COVID-19 waves, acute respiratory failure, continuous positive airway pressure, comorbidities, in-hospital mortality, Intermediate Care Unit

## Abstract

Background: In COVID-19 patients non-invasive-positive-pressure-ventilation (NIPPV) has held a challenging role to reduce mortality and the need for invasive mechanical ventilation (IMV). The aim of this study was to compare the characteristics of patients admitted to a Medical Intermediate Care Unit for acute respiratory failure due to SARS-CoV-2 pneumonia throughout four pandemic waves. Methods: The clinical data of 300 COVID-19 patients treated with continuous positive airway pressure (CPAP) were retrospectively analysed, from March-2020 to April-2022. Results: Non-survivors were older and more comorbid, whereas patients transferred to ICU were younger and had fewer pathologies. Patients were older (from 65 (29–91) years in I wave to 77 (32–94) in IV, *p* < 0.001) and with more comorbidities (from Charlson’s Comorbidity Index = 3 (0–12) in I to 6 (1–12) in IV, *p* < 0.001). No statistical difference was found for in-hospital mortality (33.0%, 35.8%, 29.6% and 45.9% in I, II, III and IV, *p* = 0.216), although ICU-transfers rate decreased from 22.0% to 1.4%. Conclusions: COVID-19 patients have become progressively older and with more comorbidities even in critical care area; from risk class analyses by age and comorbidity burden, in-hospital mortality rates remain high and are thus consistent over four waves while ICU-transfers have significantly reduced. Epidemiological changes need to be considered to improve the appropriateness of care.

## 1. Introduction

Coronavirus disease 2019 (COVID-19) has overwhelmed national healthcare systems worldwide since approximately 15% of SARS-COV-2-infected people developed severe disease that required oxygen support and 5% had a critical disease involving complications such as respiratory failure, acute respiratory distress syndrome (ARDS), sepsis and septic shock, thromboembolism, and/or multiorgan failure, including acute kidney injury and cardiac injury [1].

These proportions can be influenced by surveillance strategies, therapies and other interventions, as vaccination, regional variance in demographics and evolving variants. It is complex to determine whether a Variant of Concern causes more severe disease or higher mortality, as many other factors, as the population at risk, vaccination rates, stress on the health care systems and medical countermeasures could impact clinical outcomes.

Risk factors for deterioration, severe disease, and/or increased mortality have been widely described and include older age (>60 years), male gender and underlying non-communicable diseases. The early identification of patients at risk for and with severe disease allows for rapid referral to a designated setting in the COVID-19 care pathway with access to advanced oxygen/ventilatory support.

High flow nasal oxygen (HFNO), continuous positive airway pressure (CPAP) or non-invasive positive pressure ventilation (NIPPV) have been used in selected patients outside the intensive care units (ICUs) to reduce the need for endotracheal intubation (ETI) and invasive mechanical ventilation (IMV) [2,3,4,5,6,7,8,9,10,11], although in pre-COVID-19 era NIPPV guidelines make no recommendation on their use in ARF due to pandemic viral illnesses (referring to studies of SARS, MERS and pandemic influenza); risks include viral spread, delayed intubation, large tidal volumes and injurious transpulmonary pressures.

Non-invasive respiratory systems (NIRS) can improve oxygenation and reduce the need for IMV by increasing functional residual capacity, reducing work of breathing, and recruiting nonaerated alveoli but clinical response is difficult to be predicted by changes in commonly used gas exchange indexes [12] and by previously validated prognostic scores. Retrospective ICUs data across 2020–2021 report that NIRS have been increasingly used and independently associated with improved survival, HR 0.59 [95% CI 0.54–0.65], *p* < 0.001) [13]. Concurrently, a retrospective analysis on SARS-CoV-2 patients hospitalized in Internal Medicine wards who required oxygen support reported that, with different states of O_2_ supplementation during their hospital stay, mortality was almost exclusively associated with the use of HFNC or CPAP [14].

The use of NIRS outside the ICUs, in an appropriate setting and with appropriate monitoring, as in Intermediate Care Units (ImCUs), has become common, with a predominant use of CPAP and a PaO_2_/FiO_2_ < 150 independently associated with increased risk of failure [15], but studies show a wide percentage of success/unsuccess and propose a wide range of algorithms for starting NIRS and escalation of support to guide decision-making processes.

The principal outcomes considered are mortality, ETI for IMV and duration of non-invasive support, however different ICU admission or ETI criteria, including do-not-resuscitate orders, different definitions of respiratory deterioration or NIRS failure and different settings of application (Emergency Departments, general inpatients wards, ImCUs or ICUs) could explain some controversies observed when comparing different pandemic surges.

Few studies have, to date, compared more than two cohorts of patients with severe or critical disease focusing exclusively on those receiving CPAP support and considering the epidemiological changes, the available data regarding SARS-CoV-2 variants involved and the evolution of therapeutic strategies (including the vaccine campaign).

The aim of this study is to compare the clinical characteristics and outcomes (in-hospital mortality and ICU transfers rate for IMV) of a cohort of COVID-19 patients admitted to an out-of-ICU Medical Intermediate Care Unit for acute respiratory failure due to SARS-CoV-2 pneumonia treated with CPAP during four waves of pandemic (from March 2020 to April 2022).

## 2. Materials and Methods

In this retrospective, single-centre, non-interventional observational study all patients admitted to the High Care Internal Medicine Unit of a tertiary care hospital (Foundation IRCCS Ca’ Granda Ospedale Maggiore Policlinico, Milan, Italy) who met the inclusion/exclusion criteria during the study period were enrolled.

The inclusion criteria were: hospitalized adult patients (age > 18 years) admitted to the High Care Internal Medicine Unit for ARF due to COVID-19 pneumonia and treated with CPAP; ARF defined by an arterial partial pressure of oxygen to inspiratory oxygen fraction ratio (PaO₂/FiO₂) < 300 mmHg or by an arterial partial pressure of oxygen (PaO_2_) < 60 mmHg; pneumonia defined by chest X-ray or computed tomography with COVID-19 related pulmonary interstitial thickenings; SARS-CoV-2 infection confirmed by PCR assay for nasopharyngeal swab specimens.

Negative SARS-CoV-2 patients, positive SARS-CoV-2 patients treated with CPAP with acute respiratory failure not caused by SARS-CoV-2 interstitial pneumonia and patients with respiratory failure due to SARS-CoV-2 interstitial pneumonia treated with CPAP not in the acute phase (patients needing CPAP in the weaning phase from invasive ventilation) were excluded.

We captured data retrospectively from electronic medical records. The study collected epidemiological and medical history data, SARS-CoV-2 variants and vaccination status (when available), days since symptoms’ onset, vital signs and laboratory test at admission, arterial gas blood analysis (ABG) before CPAP, secondary diagnoses and treatments. The National Early Warning Score (NEWS), the Sequential Organ Failure Assessment (SOFA), the Acute Physiology and Chronic Health Evaluation (APACHE II) and the Charlson’s Comorbidity Index (CCI) were calculated according to standard formulae.

### Statistical Analyses

The sample’s characteristics were first presented using standard descriptive statistical analyses. Categorical variables were expressed as frequencies and percentages, while continuous variables are reported as median (min-max) according to the results of the Kolmogorov-Smirnov normality test. For categorical variables, a comparative analysis for detecting significant differences between groups was carried out using the chi-square test or Fisher’s exact test. For continuous variables, the non-parametric Mann-Whitney U-test was used. The optimal cut-offs to convert continuous variables (age and CCI) to factors were identified by maximizing Youden’s index, assessed by sensitivity and specificity. A *p*-value < 0.05 was considered statistically significant for all statistical tests.

All statistical analyses were performed using R Software with the additional package “Rcmdr”, version 2.7-1, accessed on 2 January 2023 (A language and environment for statistical computing. R Foundation for Statistical Computing, Vienna, Austria—https://www.R-project.org, accessed on 8 February 2023).

## 3. Results

According to inclusion/exclusion criteria, 300 consecutive patients were enrolled, 91 (30.3%) admitted during the first wave (1 March–31 July 2020), 81 (27%) during the second (1 August 2020–31 January 2021), 54 (18%) during the third (1 February 2021–30 April 2021) and 74 (24.7%) during the fourth wave (1 November 2021–30 April 2022).

Analysing the in-hospital patients’ flow, 256/300 patients (85.3%) were directly admitted in the High Care Internal Medicine Unit from the Emergency Department (ED), 33/300 (11%) from other inpatient wards for clinical instability and need of CPAP support and 11/300 (3.7%) from other hospitals; 110/300 (36.7%) patients were directly discharged, 68/300 (22.7%) were transferred to non-ICU wards; the ICU transfer rate was 10.7% (32/300) and the overall in-hospital mortality 36.3% (109/300).

In overall population 198/300 (66%) patients were male, the median age was 73 (29–94) years and the median time from symptom onset was 7 (1–18) days.

The comparisons of outcomes, clinical characteristics, medical history, vital signs, prognostic scores, laboratory and arterial gas blood analyses at admission throughout the four waves are reported in Table 1.

Patients were older (from 65 years (29–91) during the I wave to 77 years (32–94) during the IV, *p* < 0.001), with a significantly higher burden of comorbidities (from CCI = 3 (0–12) during the I wave to CCI = 6 (1–12) during the IV, *p* < 0.001) and showed a more frequent immunocompromised status due to diseases or medications (from 1/91, 1.1%, to 19/74, 25.7%, *p* < 0.001).

No relevant changes were found in vital signs and in arterial blood gas analyses at admission; considering the severity systems, only the APACHE II (including also chronic organ insufficiency in its score) showed differences from the I to the IV wave. Patients hospitalised during the I wave had a lower count of white blood cells with higher ferritin and CRP levels, suggesting a major inflammatory pattern in those first patients.

The viral variant detection was carried out since March 2021 (data available in 87/128 patients): Alpha B.1.1.7 was dominant during the III wave (37/40, 92.5%), Delta B.1.617.2 (23/47, 48.9%) and Omicron B.1.1.529 (24/47, 51.1%) during the first and second period of the IV wave, respectively.

In accordance with the nationwide vaccination campaign, 5/54 (9.3%) patients hospitalised during the III wave had been given the first dose of SARS-CoV-2 vaccine and 5/54 (9.3%) two doses, while 39/74 (52.7%) patients hospitalised during the IV had performed the vaccine cycle with one (3/74, 7.9%), two (19/74, 50.0%) or three doses (16/74, 42.1%).

According to local protocols and the literature’s guidance, nearly all patients have received steroids while few were eligible for antiviral or monoclonal therapy due to the severity of respiratory failure; the anti-IL1 anakinra was applied in 44/91 (48.4%) patients during the I wave while the anti-IL6 tocilizumab in two patients during the I and three patients during the IV wave, a low proportion explained by the high risk of bacterial superinfection. The antibiotics utilization gradually decreased (from 76/91, 83.5% in the I wave to 46/74, 62.2% in the IV) but remained above the percentage of bacterial infections, despite the evidence against their indiscriminate use. There has also been a progressive reduction in the cmH_2_O applied at the start of CPAP (from 10 (5–15) to 7.5 (5–12), *p* < 0.001) and in the length of treatment, but not in the overall length of stay over time.

During the I wave 20/91 (22.0%) patients were transferred to the ICU, reduced to only one/74 patient (1.4%) during the IV, *p* < 0.001; the in-hospital mortality rate, instead, was 33.0% (30/91) in the I wave, 35.8% (29/81) in the II, 29.6% (16/54) in the III and 45.9% (34/74) in the IV, without statistical differences among the four waves (*p* = 0.216), [Table 1].

Considering these primary outcomes, in overall population, survivors were younger than non-survivors (67 (30–94) vs. 78 (29–93), *p* < 0.001) and had a considerably lower CCI (3 (0–12) vs. 6 (0–12), *p* < 0.001), while patients non-transferred to ICU vs. transferred were older (74 (30–94) vs. 63.5 (29–75), *p* < 0.001) and had significantly higher CCI (5 (0–12) vs. 2 (0–5), *p* < 0.001)

These “pre-COVID-19” non-modifiable patients’ characteristics (age and medical history) were two factors significantly associated with in-hospital mortality, OR 7.03 (95%CI 3.93–12.6, *p* < 0.0001) for age ≥ 71 years and OR 10.70 (95%CI 6.03–18.90, *p* < 0.0001) for CCI ≥ 5, and were the more changing variables across the four waves.

To further highlight the epidemiological changes throughout the four waves, in Table 2 are reported the percentages of patients ≥ 71 years old and with a CCI ≥ 5, both higher in the IV wave; comparing homogeneous categories according to age and CCI, no significant differences were found in in-hospital mortality over time.

## 4. Discussion

In this selected cohort of 300 COVID-19 patients treated with CPAP in a Medical Intermediate Care Unit, non-survivors were older and more comorbid, whereas patients transferred to ICU were younger and had fewer pathologies. From the I to the IV pandemic wave (from March-2020 to April-2022) patients were older (from 65 (29–91) years in I wave to 77 (32–94) in IV, *p* < 0.001) and with more comorbidities (from CCI = 3 (0–12) in I to 6 (1–12) in IV, *p* < 0.001). No statistical difference was found for in-hospital mortality (33.0%, 35.8%, 29.6% and 45.9% in I, II, III and IV, *p* = 0.216) and, in risk class analyses by age and comorbidity burden, its rate remained high and was thus consistent over four waves while ICU-transfers were significantly reduced (from 22.0% to 1.4%).

Since the beginning of the pandemic, significant efforts have been made worldwide to identify factors associated with the increased likelihood of hospitalization and intensive treatments, defining a phenotype of patients, with older age and more comorbidities, at higher risk of developing severe respiratory failure and mortality.

Mortality data on pandemic onset were affected by the unexpected stress to which health care systems were exposed with the outbreak of SARS-CoV-2, which has often highlighted the inadequacy of adaptation to the exponential demand for care of critical patients and provided an increasing incentive for the use of NIRS in settings other than ICUs.

The first observational studies in China, Italy and the United States indicated mortality rates ranging from 12% to 28% among overall COVID-19 hospitalized patients and going as high as 49% among those admitted to ICUs, with the highest rates being observed among older patients and those with underlying pathologies [16,17,18,19].

COVID-19 in-hospital mortality then progressively declined, due to the improvements in hospital organization that allowed a better management of patient surges (i.e., expanded COVID-19 wards, increased supplies of ventilators and other critical equipment); earlier hospitalization as a result of easier access to testing; and reinforcements in COVID-19 treatments [20,21,22,23].

These management remarks can be a first explanation to the reduction of adverse outcomes after the very early pandemic stage; since Italy was the first Western nation in which the pandemic spread, the lower age and lower number of comorbidities found in our I wave patients may be a consequence that those older and more compromised were initially widely distributed in less-intensive hospital settings despite critical illness.

A prospective cohort study analysed the time-dependent probability of death in patients admitted to a COVID-19 referral centre in Milan, Italy, from February 2020 to April 2021, reporting that hospitalization during the II and the III wave was independently associated with a significantly lower risk of death but, among the patients aged > 75 years, there was no significant difference during the three waves [24].

Beyond that, the epidemiological transition of hospitalised patients should be considered analysing the subsequent waves.

The analyses of health-administrative data of 4 million inhabitants in North-West Italy during three pandemic waves showed in the II and III a reduction in median age, comorbidity burden, mortality in outpatients, inpatients, and patients admitted to ICUs and IMV but a parallel increase in the use of CPAP, confirming a general trend towards younger and healthier patients over time but also showed an independent effect of the period on mortality and ICU admission: new viral variants, the starting of vaccination, organizational improvements in tracking, outpatients and inpatients management could have influenced these trends [25].

In different countries, a shift has been reported among patients infected with SARS-CoV-2 towards those younger and with fewer comorbidities, with a lower proportion of patients requiring oxygen therapy, a shorter length of hospital stay and lower mortality risk as the pandemic evolved. Most of these studies, however, include case-mixes from medical wards and critical care units, where the range of fatality rates was considerably different and thereby potential selection biases could have been present [26,27,28,29,30,31,32,33].

Data from a prospective study from March 2020 to June 2021 reported that the overall in-hospital mortality rate decreased during the first three waves and fell from 18% to 4%, with a comparable NEWS2 score upon admission but with a progressive lower CCI and age [34].

According to our results, other studies showed that among the I and II wave hospitalised COVID-19 patients became progressively older and with a higher prevalence of all underlying diseases, without differences in symptoms, vital signs, laboratory findings, and blood gases at the time of admission, with a reduced proportion of ICUs admissions, but contrasting results in survival prognoses [35,36,37,38,39].

In our study, considering lymphocytes’ count, CRP and ferritin, a decreasing trend in the inflammatory burden has been noted from the first to the fourth wave, while vital signs, blood gases, NEWS and SOFA upon admission were comparable throughout all four waves.

A multicentre study between February 2020 and March 2021 reported that, in ICUs, severity scores were lower in the II/III waves compared to the I wave according with APACHE II (12 [IQR 916] vs. 14 [IQR 1019]) and SOFA (4 [IQR 36] vs. 5 [IQR 37], *p* < 0.001), with fewer differences in ICUs mortality rates (I wave 31.7% vs. II/III waves 28.8%, *p* = 0.06) [40].

We reported similar results with a median APACHE II = 12 (2–18) and SOFA = 3 (2–5) for patients transferred to ICU due to CPAP failure, without differences overtime; in this sub-cohort the overall in-hospital mortality was 37.5% (12/32), all patients underwent ETI for mechanical ventilation, no one received ECMO, but we are unable to report the specific exclusion criteria.

Furthermore, in our homogeneous by severity of illness cohort, it was confirmed that older age and higher CCI independently predispose to a significantly higher risk of in-hospital mortality.

From risk class analyses by age and comorbidity burden the in-hospital mortality rate was thus consistent across the four waves suggesting that once severe SARS-CoV-2 illness develops, mortality is definitely high especially in complex, comorbid and immunocompromised COVID-19 patients.

Agreeing that frailty is highly prevalent among COVID-19 CPAP patients and predicts poorer outcomes, also independently of age, a personalization of care balancing the risk and benefit of treatments (especially the invasive ones) in such complex patients is pivotal [41]; it has already been proven that the proportion of IMV decreased with increasing age and frailty [42].

The marked reduction observed in our cohort in transfers to ICU for IMV from the first to the fourth wave did not affect the in-hospital mortality and can be traced back to the goal of an appropriate selection of eligible patients for escalation of care intensity (i.e., ETI and IMV), with age and co-morbidities as the main evaluation criteria.

The main limitations of this study consisted in its retrospective and monocentric design. All data considered were those available at the time of admission to the High Care Unit, before starting CPAP support, with an immutable risk of missing data. It has not been possible to establish standard treatments, parameters, and duration of CPAP, over time adapted to the evolution of international recommendations, local protocols and patients’ complexity. Moreover, all patients admitted during the study period, as reported by electronic reports after appropriate query, have been enrolled in accordance with the inclusion/exclusion criteria, but the single-centre (single-ward) design, without internal/external control groups, may have conditioned an inflow selection bias related to the changing macro-organization of the hospital which, in the case of the COVID-19 hub in Milan, underwent to major management changes during the four waves in the number of COVID-19 beds and sub-intensive areas for CPAP treatment.

Thus, this study only served descriptive and exploratory purposes and our findings should be confirmed in larger multicentre studies.

## 5. Conclusions

Comparing the clinical characteristics of COVID-19 patients with severe or critical illness treated with CPAP, the in-hospital mortality rate remains high. From a similar severity of respiratory failure over the four waves, the most significant differences were found in the characteristics of the hospitalised population, which, even in the critical care area, has become progressively older, frailer and more comorbid.

To date, few analyses have included and compared multiple pandemic waves focusing on a specifically selected cohort who required CPAP support. Nearly three years after the first COVID-19 patient in Italy, this study emphasizes the strict need for an adequate profiling of clinical risk, considering epidemiological changes and tailoring treatments on each single patient to improve the appropriateness of care, mainly invasive supports.

## Figures and Tables

**Table 1 jcm-12-01562-t001:** Comparison of waves: clinical characteristics, anamnestic data, vital signs, prognostic scores, laboratory and arterial blood gas analyses at admission.

	Overall (300)	I (91)	II (81)	III (54)	IV (74)	*p*-Value
Age	73 29–94	65 29–91	75 39–94	71 31–86	77 32–94	<0.001
Male	198 (66.0%)	63 (69.2%)	60 (74.1%)	34 (63.0%)	41 (55.4%)	0.082
Mortality	109 (36.3%)	30 (33.0%)	29 (35.8%)	16 (29.6%)	34 (45.9%)	0.216
ICU transfers	32 (10.7%)	20 (22.0%)	4 (4.9%)	7 (13.0%)	1 (1.4%)	<0.001
LoS tot (days)	19.5 1–122	24 2–105	21 1–122	18.5 2–85	17 1–90	0.087
Los CPAP (days)	8 1–96	8.5 1–96	10 1–37	7 1–37	7 1–32	0.026
FiO_2_ CPAP (%)	0.50 0.30–0.95	0.50 0.35–0.90	0.60 0.40–0.80	0.50 0.40–0.80	0.60 0.30–0.95	0.188
PEEP CPAP (cmH_2_O)	7.5 5–15	10 5–15	7.5 5–12.5	7.5 5–11	7.5 5–12	<0.001
Symptom Onset (days)	7 1–18	7 1–18	7 1–15	7 1–15	5 1–16	0.280
Hypertension	161 (53.7%)	37 (40.7%)	47 (58.0%)	30 (55.6%)	47 (63.5%)	0.021
CVD/HF	80 (26.7%)	20 (22.0%)	24 (29.6%)	10 (18.5%)	26 (35.1%)	0.115
COPD	61 (20.3%)	13 (14.3%)	17 (21.0%)	9 (16.7%)	22 (29.7%)	0.087
Diabetes	62 (20.7%)	19 (20.9%)	22 (27.2%)	8 (14.8%)	13 (17.6%)	0.302
Renal Failure	36 (12.0%)	10 (11.0%)	12 (14.8%)	2 (3.7%)	12 (16.2%)	0.141
Liver Disease	8 (2.7%)	2 (2.2%)	2 (2.5%)	4 (7.4%)	0 (0.0%)	0.079
Cancer	40 (13.3%)	7 (7.7%)	11 (13.6%)	6 (11.1%)	16 (21.6%)	0.068
Rheumatological Disease	28 (9.3%)	6 (6.6%)	8 (9.9%)	4 (7.4%)	10 (13.5%)	0.457
Cerebral Vasculopathy	55 (18.3%)	13 (14.3%)	19 (23.5%)	7 (13.0%)	16 (21.6%)	0.262
Immunodepression	30 (10.0%)	1 (1.1%)	8 (9.9%)	3 (5.6%)	19 (25.7%)	<0.001
N. diseases	2 0–8	1 0–5	2 0–8	1 0–7	3 0–7	<0.001
N. diseases ≥ 2	162 (54.0%)	39 (42.9%)	48 (59.3%)	25 (46.3%)	50 (67.6%)	0.007
CCI	4 0–12	3 0–12	5 0–12	4 0–11	6 1–12	<0.001
SpO2 (%)	95 70–100	96 70–100	95 75–100	95 90–100	95 78–100	0.273
RR (breaths/min)	24 12–40	25 16–40	24 16–40	24 12–40	24 14–40	0.162
HR (beats/min)	82 50–170	87 50–116	80 56–135	78 54–130	80 50–170	0.027
BP dias (mmHg)	80 40–115	80 50–115	80 50–111	80 50–110	77 40–100	0.615
BP sys (mmHg)	135 70–200	135 95–175	140 100–200	140 90–180	137 70–180	0.241
Temperature (°C)	36.0 35.0–40.0	37.0 35.0–40.0	36.0 36.0–39.7	36.0 36.0–38.5	36.0 36.0–38.2	<0.001
GCS	15 7–15	15 12–15	15 13–15	15 13–15	15 7–15	0.519
NEWS	5 0–15	5 0–15	5 2–13	5 2–11	6 2–13	0.276
SOFA	3 1–10	4 1–10	3 1–9	3 2–7	4 1–9	0.381
APACHE II	14 0–33	12 0–33	15 0–27	13 3–23	15 2–30	<0.001
WBC (10^9^/L)	7.8 0.6–72.9	6.8 0.9–22.6	9.3 1.0–26.1	7.0 2.4–17.6	8.2 0.6–72.9	<0.001
Lymphocytes (10^9^/L)	0.8 0.1–7.6	0.9 0.2–3.5	0.8 0.2–7.6	0.7 0.1–2.4	0.8 0.2–1.8	0.073
Hb (g/dL)	13.0 7.4–17.2	12.8 8.7–16.7	13.0 8.4–16.7	12.7 7.4–16.3	13.2 7.9–17.2	0.918
Platelets (10^9^/L)	232.5 25–639	228 60–584	257 25–620	222 99–487	232 45–639	0.342
Urea (mg/dL)	44 10–414	37 11–162	49 19–414	42 10–127	54 18–235	<0.001
Creatinine (mg/dL)	0.97 0.38–7.32	0.94 0.38–5.91	1.03 0.58–7.32	0.88 0.46–1.99	0.95 0.50–5.50	0.012
AST (UI/L)	64 10–720	68.0 24.0–720.0	56.0 24.0–144.0	55.5 20.0–194.0	68.0 10.0–594.0	0.486
D-dimer (mg/L)	1.1 0.2–247.3	1.06 0.20–109.40	1.26 0.25–247.30	1.01 0.31–137.24	1.36 0.20–106.64	0.217
LDH (UI/L)	351 133–2475	341 170–803	344 133–2475	370 182–776	359 140–998	0.358
Ferritine (mcg/L)	961 24–10,211	1408 225–10,211	884 51–4571	746 24–7883	880 38–5579	0.002
CRP (mg/dL)	8.7 0.2–45.2	12.13 0.89–45.22	8.10 0.19–32.52	6.77 0.71–27.80	7.25 0.27–34.76	<0.001
PCT (ng/mL)	0.19 0.02–56.7	0.25 0.07–6.85	0.15 0.02–27.10	0.14 0.03–10.30	0.17 0.03–56.70	0.126
pH	7.48 6.95–7.70	7.47 6.95–7.58	7.48 7.31–7.62	7.49 7.09–7.59	7.48 7.28–7.70	0.176
PaO_2_ (mmHg)	67 25–189	69 36–189	67 28–142	69 33–115	65 25–114	0.234
PaCO_2_ (mmHg)	33 17–76	34 17–49	32 20–66	33 23–76	33 19–69	0.748
Lactate (mmol/L)	1.4 0.4–8.0	1.2 0.4–8.0	1.4 0.6–5.4	1.4 0.4–7.8	1.4 0.5–5.6	0.030
HCO_3_ (mmol/L)	25.0 9.9–38.0	25.0 9.9–32.6	25.1 14.0–37.2	24.7 14.6–31.3	25.6 10.3–38.0	0.717
PaO_2_/FiO_2_	134.0 47.8–278.6	152.5 50.0–278.6	133.3 50.0–214.3	142.0 66.7–276.2	119.5 47.8–276.2	0.073

LoS: Length of Stay; CVD: cardiovascular disease; HF: heart failure; COPD: chronic obstructive pulmonary disease; CCI: Charlson Comorbidity Index; RR: respiratory rate; HR: heart rate; BP dias: diastolic blood pressure; BP sys: systolic blood pressure; GCS: Glasgow Coma Scale; NEWS: National Early Warning Score; SOFA: Sequential Organ Failure Assessment; APACHE II: Acute Physiology and Chronic Health Evaluation; CRP: C Reactive Proteine; PCT: Procalcitonine; PaO_2_: arterial partial pressure of oxygen; PaCO_2_: arterial partial pressure of carbon dioxide; FiO2: fraction of inspired oxygen.

**Table 2 jcm-12-01562-t002:** In-hospital mortality stratified by age and CCI: comparison of waves.

WAVE	I (91)	II (81)	III (54)	IV (74)	*p*-Value
AGE ≥ 71 172 (57.3%)	37 (40.7%)	57 (70.4%)	29 (53.7%)	49 (66.2%)	<0.001
CCI ≥ 5 143 (47.6%)	30 (33%)	44 (54.3%)	21 (38.9%)	48 (64.9%)	<0.001
AGE < 71					
Non-Survivors	9 (16.7%)	1 (4.2%)	3 (12.0%)	5 (20.0%)	0.382
AGE ≥ 71					
Non-Survivors	21 (56.8%)	29 (50.9%)	13 (44.8%)	29 (59.2%)	0.609
CCI < 5					
Non-Survivors	9 (14.8%)	3 (8.1%)	5 (15.2%)	4 (15.4%)	0.761
CCI ≥ 5					
Non-Survivors	21 (70%)	27 (61.4%)	11 (52.4%)	30 (62.5%)	0.647

## Data Availability

The data presented in this study are available on request from the corresponding author.

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
