# Peer review of "Clinical Characteristics and Outcomes of Patients with Acute Respiratory Failure Due to SARS-CoV-2 Interstitial Pneumonia Treated with CPAP in a Medical Intermediate Care Setting: A Retrospective Observational Study on Comparison of Four Waves"

_jcm, 2023, doi:10.3390/jcm12041562_

Round 1
Reviewer 1 Report
This is a very interesting study concerning the outcomes of Covid-19 ARDS patients treated with CPAP. The authors retrospectively analysed 300 patients and tried to identify predictors of survivla ad intubation in the cohort covering all four waves. Although there are interesting data in the study, the manuscript needs extensive modifications before it could be reconsidered.
the presentation of the results is not so clear. Please replace no-survivors to non-survivors and present in groups of survivors and non-survivors the results without confusing the order of presentation.
what does anamnestic mean?
Did the patients receive anti-IL6 ? as by recommendation?
what was the outcome of patients rceiving anti-IL1, which was later considered in the treatent regimen?
in the tables explain all the abbreviated words.
Linf means Lymphocytes?
The discussion needs re-arrangement as it needs to present your findings and compare/discuss them with ulished data.
the power analysis is not well presented.
Author Response
REVIEWER 1
Point 1: “Please replace no-survivors to non-survivors and present in groups of survivors and non-survivors the results without confusing the order of presentation”
*Response 1:
In the results section we have changed definition and order for survivors and non-survivors as suggested.
Point 2: “what does anamnestic mean?”
*Response 2:
“Anamnestic” is the adjective related to the medical history of a patient, i.e. the anamnesis.
Point 3: “Did the patients receive anti-IL6? as by recommendation? What was the outcome of patients receiving anti-IL1, which was later considered in the treatment regimen?”
*Response 3:
We reported in the tables below the available data regarding the main involved drugs, including anti-IL6 and anti-IL1; according to recommendation all CPAP patients received steroids, few were eligible for antiviral or monoclonal therapy due to the severity of respiratory failure, and a non-significant proportion could be treated with immunomodulators especially considering the high risk of bacterial superinfection.
|
|
I wave (91) |
II wave (81) |
III wave (54) |
IV wave (74) |
p-value |
|
Remdesivir |
11 (12.1%) |
2 (2.5%) |
6 (11.1%) |
18 (24.3%) |
0.001 |
|
Monocl. Ab |
0 (0.0%) |
0 (0.0%) |
0 (0.0%) |
3 (4.1%) |
0.026 |
|
Anakinra |
44 (48.4%) |
0 (0.0%) |
0 (0.0%) |
0 (0.0%) |
< 0.001 |
|
Tocilizumab |
2 (2.2%) |
0 (0.0%) |
0 (0.0%) |
3 (4.1%) |
0.170 |
|
Antibiotics |
76 (83.5%) |
68 (84.0%) |
33 (61.1%) |
46 (62.2%) |
< 0.001 |
|
|
Tot (300) |
Survivors (191) |
Non-survivors (109) |
p-value |
|
Remdesivir |
37 (12.3%) |
27 (14.1%) |
10 (9.2%) |
0.091 |
|
Monocl. Ab |
3 (1%) |
1 (0.5%) |
2 (1.8%) |
0.299 |
|
Anakinra |
44 (14.7%) |
34 (17.8%) |
10 (9.2%) |
0.043 |
|
Tocilizumab |
5 (1.67%) |
2 (1%) |
3 (2.8%) |
0.357 |
Point 4: “In the tables explain all the abbreviated words. Linf means Lymphocytes?”
*Response 4:
We have reposted all abbreviations at the bottom of the tables and corrected the reported misprint.
Point 5: “The discussion needs re-arrangement as it needs to present your findings and compare/discuss them with published data.”
*Response 5:
We have uploaded a revised manuscript according to all reviewers’ comments.
Point 6: “the power analysis is not well presented.”
*Response 6:
This study has a descriptive and exploratory purpose and given the retrospective design its sample size depends solely on the actual number of patients who fell within the inclusion criteria during the study period. Based on the selected outcomes, considering data of the main studies reported in literature and cited in the paper with a mortality rate varying between 15%-30%and a CPAP failure coming up to 45%, we calculated the sample size and relative power of the analyses only to have a minimum cut off of patients to be included assuming the risk of missing data. By accepting a type I error of 0.05 (a = 0.05), with a study population of 200 patients a power of 80% (1- type II error, b = 0.2) is already been obtained.

Reviewer 2 Report
The study should focus on its primary aim "to compare the characteristics of patients admitted to a Medical-Intermediate-Care-Unit for acute respiratory failure due to SARS-CoV-2 pneumonia throughout four pandemic waves" and not repeat anapysis for factors affecting survival, which has been repeated in much larger cohorts. In that perspective only Table 3 and 5 are relevant. The rest should be omited. Discussion should also be limited and more focused to comparing the four waves. Conclusion should be shorten too.
Author Response
REVIEWER 2
Point 1: “The study should focus on its primary aim and not repeat analysis for factors affecting survival, which has been repeated in much larger cohorts. In that perspective only Table 3 and 5 are relevant. The rest should be omited. Discussion should also be limited and more focused to comparing the four waves. Conclusion should be shorten too.”
*Response 1:
We have uploaded a revised manuscript according to all reviewers’ comments.

Reviewer 3 Report
Major
Severe acute respiratory syndrome-related coronavirus 2 (SARS-CoV-2), which causes the coronavirus disease 2019 (COVID-19), results in severe acute respiratory failure, and death. While the fundamental principles of acute respiratory failure management are similar between COVID-19 and non-COVID-19 patients, there are some notable differences, including a focus on provider safety. However, more than three years experiences of CIVID-19 pandemic change the treatment strategy and disease severity of COVID-19 itself.
The current study was conducted to compare the characteristics of patients admitted to a Medical-Intermediate-Care-Unit for acute respiratory failure due to SARS-CoV-2 pneumonia throughout four pandemic waves. Clinical data of 300 COVID-19 patients treated with continuous positive airway pressure (CPAP) have been collected and analysed, from March-2020 to April-2022. The data indicated that no-survivors were older and more comorbid, whereas patients transferred to ICU were younger and had fewer pathologies. No statistical difference was found for in-hospital mortality, although ICU-transfers rate decreased from 22.0% to 1.4%. In their conclusions: COVID-19 patients have become progressively older and more complex even in critical care area; from risk class analyses by age and comorbidity burden, in-hospital mortality rate remains high and is thus consistent over four waves. The data are pretty much convincing , and informative for many clinicians for treating the COVID-19 in clinical settings,.
However the data are lacking of mechanical ventilation and ECMO indication for COVID-10 patients in particular for older cases aged more than 70years.
In this point,
Two papers should be cited.
An important paper such as COVID-‘21’ pandemic is very different from COVID-19 pandemic in Japan - A special focus on the frequency of mechanical ventilation and ECMO treatments, by Teramoto S(Open Journal of Public Health 4(3) 1037-1038 202)
Teramoto S (2020) Japan famous comedian with COPD was killed by SARS-CoV-2-caused pneumonia –implication for smoking risk and ECMO indication. Int Clin Med 4: DOI: 10.15761/ICM.1000180.
Author Response
REVIEWER 3
Point 1: “the data are lacking of mechanical ventilation and ECMO indication for COVID-10 patients in particular for older cases aged more than 70 years. Two papers should be cited:
An important paper such as COVID-21 pandemic is very different from COVID-19 pandemic in Japan - A special focus on the frequency of mechanical ventilation and ECMO treatments, by Teramoto S (Open Journal of Public Health 4(3) 1037-1038 202)
Teramoto S (2020) Japan famous comedian with COPD was killed by SARS-CoV-2-caused pneumonia –implication for smoking risk and ECMO indication. Int Clin Med 4: DOI: 10.15761/ICM.1000180.”
*Response 1:
We thank the reviewer for this discussion topic, we agree that ECMO has emerged as a viable option in critically ill COVID-19 patients who need more support than invasive mechanical ventilation. Particularly during pandemic, eligibility criteria should consider both the extensive human, technical and financial resource consumption, and clinical indicators of effectiveness so that older patients with COPD - as well as with other chronic diseases - may be excluded (both factors independently associated with an increased mortality risk).
The focus of this study was to describe the clinical characteristics of a cohort of critical COVID-19 patients in an Intermediate Care setting; re-analyzing electronic clinical records, patients transferred to ICU due to CPAP failure were all treated with mechanical ventilation, no one with ECMO, but we are unable to report the specific exclusion criteria.
We have uploaded a revised manuscript according to all reviewers’ comments.

Round 2
Reviewer 1 Report
The authors have provided an improved manuscript.
Yet there are some points that need further clarification.
1. a regression analysis to evaluate factors affecting mortality would be useful in the whole cohor and everey wave separately
2. in the abstract replace no-survivors. I think "anamnestic" should be replaced with medical history, "complex" patients, with patients with more comorbidities.
3. the last sentence in the abstract is not supported by the findings
4. in the reults each paragraph should not be started with a number
5. line 140 change "for" to "four"
6. the first paragraph of the discussion should summarise the findings of the study and then each paragraph hould analyse the findings and compare them to the published data.
7. lines 270-272. the meaning is difficult to follow
8. in the conclusion you refer to the high case mix. How is this supported from the findings of the study?
Author Response
Point 1: “a regression analysis to evaluate factors affecting mortality would be useful in the whole cohort and every wave separately”
*Response 1: according to other reviewers’ comments we have focused the study on the four waves comparison and have not reported analysis for factors affecting survivals, which has been repeated in much larger cohorts. We propose here the results of a selected univariate logistic regression for in-hospital mortality, based on statistical significance, not deeming it appropriate to consider the combined outcomes (only one patient was transferred to ICU during IV wave). Cut-off values have been determined by the best Youden’s Index in the overall population. Some discrepancies observed comparing the four waves can be attributed to the small sample size of each cohort.
|
Overall population |
|||
|
|
OR |
95% CI |
p-value |
|
AGE |
1.08 |
1.06-1.11 |
< 0.0001 |
|
AGE ≥ 71 |
7.03 |
3.93-12.6 |
< 0.0001 |
|
NEWS |
1.59 |
1.40-1.82 |
< 0.0001 |
|
NEWS ≥ 6 |
6.02 |
3.75-10.20 |
< 0.0001 |
|
PaO2/FiO2 |
1.00 |
0.99-1.00 |
0.6630 |
|
PaO2/FiO2 ≤ 98 |
1.96 |
1.12-3.45 |
0.0186 |
|
CHARLSON |
1.57 |
1.40-1.77 |
< 0.0001 |
|
CHARLSON ≥ 5 |
10.70 |
6.03-18.90 |
< 0.0001 |
|
SOFA |
2.01 |
1.60-2.53 |
< 0.0001 |
|
SOFA ≥ 4 |
4.67 |
2.79-7.81 |
< 0.0001 |
|
APACHE II |
1.40 |
1.29-1.53 |
< 0.0001 |
|
APACHE II ≥ 14 |
11.70 |
6.44-21.10 |
< 0.0001 |
|
Immunodepression |
4.00 |
1.80-8.90 |
0.00068 |
|
|
I OR (95% CI) |
II OR (95% CI) |
III OR (95% CI) |
IV OR (95% CI) |
|
AGE |
1.07 (1.03-1.12) p = 0.0009 |
1.11 (1.04-1.18) p = 0.0008 |
1.09 (1.02-1.18) p = 0.0182 |
1.08 (1.02-1.14) p = 0.0063 |
|
AGE ≥ 71 |
6.56 (2.50-17.30) p = 0.00013 |
23.80 (3.01-188.0) p = 0.0027 |
5.96 (1.45-24.40) p = 0.0132 |
5.80 (1.87-18.00) p = 0.0024 |
|
NEWS |
1.42 (1.14-1.76) p = 0.0015 |
1.66 (1.28-2.16) p = 0.0001 |
1.64 (1.17-2.31) p = 0.0040 |
1.76 (1.33-2.32) p < 0.0001 |
|
NEWS ≥ 6 |
2.32 (0.82-6.54) p = 0.1120 |
6.09 (2.01-18.40) p = 0.0014 |
8.94 (1.77-45.30) p = 0.0081 |
6.77 (2.17-21.1) p = 0.0010 |
|
PaO2/FiO2 |
1.01 (1.00-1.02) p = 0.3630 |
1.01 (0.99-1.01) p = 0.8580 |
1.00 (0.98-1.01) p = 0.5000 |
1.00 (0.99-1.01) p = 0.694 |
|
PaO2/FiO2 ≤ 98 |
3.22 (1.15-9.09) p = 0.0258 |
1.96 (0.64-5.88) p = 0.2350 |
0.91 (0.16-5.29) p = 0.920 |
1.43 (0.50-4.08) p = 0.499 |
|
CHARLSON |
1.72 (1.37-2.16) p < 0.0001 |
1.85 (1.39-2.47) p < 0.0001 |
1.36 (1.04-1.77) p = 0.0240 |
1.42 (1.14-1.75) p = 0.0014 |
|
CHARLSON ≥ 5 |
13.5 (4.70-38.70) p < 0.0001 |
18.00 (4.77-67.90) p < 0.0001 |
6.16 (1.71-22.20) p = 0.0054 |
9.17 (2.72-30.90) p = 0.0003 |
|
SOFA |
2.04 (1.34-3.10) p = 0.0009 |
2.04 (1.31-3.18) p = 0.0016 |
1.97 (1.07-3.59) p = 0.0282 |
1.91 (1.24-2.93) p = 0.0031 |
|
SOFA ≥ 4 |
4.01 (1.53-10.50) p = 0.0046 |
10.60 (3.57-31.30) p < 0.0001 |
3.08 (0.91-10.40) p = 0.0702 |
3.25 (1.23-8.55) p = 0.0171 |
|
APACHE II |
1.43 (1.21-1.69) p < 0.0001 |
1.69 (1.31-2.17) p < 0.0001 |
1.38 (1.13-1.67) p = 0.0013 |
1.30 (1.11-1.51) p = 0.0008 |
|
APACHE II ≥ 14 |
17.30 (5.71-52.60) p < 0.0001 |
32.2 (4.83-303.00) p = 0.0005 |
21.80 (4.14-115.0) p = 0.0003 |
5.16 (1.75-15.2) p = 0.0029 |
|
Immunodepression |
0.00 (/-0.00) p = 0.992 |
3.20 (0.71-14.50) p = 0.131 |
0.00 (0.00-/) p = 0.993 |
3.09 (1.01-9.45) p = 0.0477 |
Point 2: “in the abstract replace no-survivors. I think "anamnestic" should be replaced with medical history, "complex" patients, with patients with more comorbidities”
*Response 2: we have uploaded a new revised manuscript according to the reviewer’s comments.
Point 3: “the last sentence in the abstract is not supported by the findings”
*Response 3: we have changed the conclusion of the abstract in accordance with the new version of the results.
Point 4: “in the results each paragraph should not be started with a number”
*Response 4: we have uploaded a new revised manuscript according to this comment.
Point 5: “line 140 change "for" to "four"
*Response 5: it has been corrected in the manuscript
Point 6: “the first paragraph of the discussion should summarise the findings of the study and then each paragraph should analyse the findings and compare them to the published data”
*Response 6: we have uploaded a new revised manuscript according to this comment.
Point 7: “lines 270-272. the meaning is difficult to follow”
*Response 7: it has been changed in the manuscript.
Point 8: “in the conclusion you refer to the high case mix. How is this supported from the findings of the study?”
*Response 8: the case-mix shift toward higher values throughout pandemic waves was captured using Diagnosis-Related-Groups coded at discharge and considering ICD-9 classification other than SARS-CoV-2 pneumonia. This further analysis would have complicated the rendering of results and was beyond the aim of the work; to avoid misunderstandings, we have removed the sentence.
Thank you to the reviewer for the comments that allowed us to improve this work
Reviewer 2 Report
I have no further comments
Author Response
Point 1: “I have no further comments”
*Response 1:
Thank you to the reviewer for the comments that allowed us to improve this work